# Microencapsulated Tuna Oil Results in Higher Absorption of DHA in Toddlers

**DOI:** 10.3390/nu12010248

**Published:** 2020-01-18

**Authors:** Samaneh Ghasemi Fard, Su Peng Loh, Giovanni M. Turchini, Bo Wang, Glenn Elliott, Andrew J. Sinclair

**Affiliations:** 1Nu-Mega Ingredients Pty Ltd., Brisbane, QLD 4000, Australia; samanehf@nu-mega.com (S.G.F.); glenne@nu-mega.com (G.E.); 2Department of Nutrition & Dietetics, Faculty of Medicine & Health Sciences, Universiti Putra Malaysia, Serdang 43300, Selangor, Malaysia; sploh@upm.edu.my; 3School of Life and Environmental Sciences, Deakin University, Geelong, VIC 3220, Australia; giovanni.turchini@deakin.edu.au; 4School of Behavioural and Health Science, Australian Catholic University, Sydney, NSW 2000, Australia; Bo.Wang@acu.edu.au; 5Department of Nutrition, Dietetics and Food, Monash University, Clayton, Melbourne, VIC 3168, Australia

**Keywords:** bioavailability, microencapsulation, tuna oil, omega-3, DHA, toddlers

## Abstract

Docosahexaenoic acid (DHA) is an essential component for brain and visual acuity development during foetal and early postnatal life. A newly released directive under the European Commission stipulates DHA as a mandatory ingredient in infant formula. This poses challenges to manufacturers in preserving the stability and bioavailability of DHA at levels akin to human breast milk. The aims of this study were (a) to investigate the bioavailability of microencapsulated omega-3 DHA formulations in healthy toddlers compared with high DHA fish oil for a one-month period and (b) to assess the effect of DHA supplementation on children’s sleep and cry patterns. Sixty toddlers were randomly allocated to four groups: 1. unfortified formula, 2. unfortified formula plus high DHA tuna oil, 3. fortified formula with dairy-based microencapsulated high DHA tuna oil powder, and 4. fortified formula with allergenic-free microencapsulated high DHA tuna oil powder. Bioavailability was assessed from both blood and faecal fatty acid levels. The results showed an enhanced bioavailability with significantly greater concentrations of blood DHA levels in formulas with microencapsulated powders. There were no significant effects of treatment on sleep and cry patterns. Application and delivery of microencapsulated DHA tuna oil powder in toddlers’ formula provided better bioavailability of the active DHA.

## 1. Introduction

Tuna oil contains high levels of omega-3 long chain polyunsaturated fatty acids (*n*-3 LC-PUFA) including docosahexaenoic acid (DHA; 22:6*n*-3), eicosapentaenoic acid (EPA; 20:5*n*-3) as well as a small quantity of docosapentaenoic acid (DPA; 22:5*n*-3). Compared with traditional fish oils originating from small pelagic fish such as anchovies, which typically contain EPA and DHA in a ratio of 18/12 (180 mg of EPA and 120 mg of DHA per g of oil), tuna oil is richer in DHA and contains less EPA (60 mg of EPA and 260 mg of DHA per g of oil). DHA plays important roles, different to those of EPA, in heart, cardiovascular, brain and visual functions [1].

The early finding that the mammalian brain is invariably rich in DHA [2] stimulated research into maternal and early infant nutrition, and it is now widely accepted that DHA is necessary for growth [3,4,5], the maturation of infants’ and toddlers’ brains [6,7,8,9,10,11] and for visual acuity [12,13,14,15,16]. Prior to birth, DHA can be provided to the foetus via placental transfer from maternal DHA stores during pregnancy. Infants and toddlers can receive DHA via maternal milk after birth, or through DHA-fortified infant formula and food [17].

During the transition from complementary feeding at 6 months of age to a mixed and varied diet at 36 months of age, it is often important to consume *n*-3 LC-PUFA and essential fatty acids at the level of dietary recommendations. However, the current median *n*-3 LC-PUFA and estimated DHA intakes in toddlers in most countries are lower than the recommended levels [18,19,20,21,22,23,24] (Appendix A). The EFSA recommend that infants and young children (<24 months) should consume 100 mg of DHA per day, while for older children (2–18 years), they recommend a daily intake of 250 mg [25]. Similarly, recommendations made by the Food and Agriculture Organization of the United Nations (FAO) and the World Health Organisation (WHO) specify that the minimum daily intake of EPA plus DHA for pregnant and lactating women is 300 mg, with at least 200 mg being DHA [26]. In order to bridge the gap between the current intake and recommended levels of *n*-3 LC-PUFA, general foods, especially infant and toddler formulas, should be enriched with *n*-3 LC-PUFA, particularly DHA.

In children, sleep problems have been associated with poor health, behavioural and cognitive problems in many cases [27]. Even though the effects of omega-3 fatty acid intake on sleep in toddlers have not been investigated in depth, a few studies showed an association between sleep disorders and low DHA status in children aged 7–9 years [28,29]. Richardson showed that children with more serious sleep problems had lower blood concentrations of DHA, and lower ratios of DHA:ARA (arachidonic acid, 20:4n-6) in particular [28]. Montgomery et al. [29] showed that DHA supplementation in children led to increased sleep duration and fewer waking episodes per night. In another study, the frequent consumption of fish was found to be independently associated with less sleep disturbances in children aged 9–11 years, which indicated better overall sleep quality [30].

To facilitate a greater intake of DHA, a variety of foods fortified with *n*-3 LC-PUFA have been developed. However, as *n*-3 LC-PUFA are prone to degradation via peroxidation, it can be challenging to fortify food products with them. Oxidative degradation leads to the formation of free radicals and reactive aldehydes with undesirable odours and adverse sensory taste [31,32]. To overcome this issue and maintain the oxidative stability of fish oils, typically additional antioxidants have been used. Alternatively, the oil can be microencapsulated to protect it from external factors and minimize the risk of peroxidation. In this way, it can also be converted into a powder form, which would further facilitate stability, handling, processing and storage [33,34]. Although a variety of encapsulation technologies such as freeze drying [35,36], spray dried emulsion [37], spray dried coacervates [38], ultrasonic atomisation [39], or a combination of these methods, have been developed to protect fish oils against oxidative deterioration. Spray dried emulsion is still the methodology most widely used, due to its lower operating costs, simplicity [40] and suitability for use within infant and toddler formula and foods due to the type of encapsulation materials used i.e., milk protein and carbohydrates [41].

Regardless of the selected technology or formulation, microencapsulated fish oils need to be adequately bioavailable, so they can be used as physiologically effective food ingredients. Several studies have investigated the bioavailability of microencapsulated high EPA fish oils using different technologies and formulations [42,43,44,45,46,47,48] (Appendix A). In particular, two studies compared the bioavailability of microencapsulated high EPA fish oil versus non-microencapsulated fish oil in adults [42,43]. One of these studies showed that microencapsulated high EPA fish oil had better bioavailability compared with non-microencapsulated fish oil [43], whereas Hinriksdottir et al. [42] showed a similar bioavailability in both groups. There is a paucity of information about bioavailability of omega-3 fatty acids in powder in children and for high DHA oils.

Despite evidence surrounding the effect of microencapsulation on the stability of fish oil in the final product, there is no evidence relative to possible effects on the efficacy of absorption of *n*-3 LC-PUFA from oils in such powders in toddlers, in comparison with fish oil. Based on this background information, the present study was developed to address two separate, but interconnected objectives. Specifically, the first objective was to investigate the bioavailability of different microencapsulation matrices for high DHA oils in healthy toddlers (1–4 years old) compared with high DHA fish oil for a period of one month. The second objective was to assess the effect of DHA supplementation on toddlers’ cry and sleep patterns.

## 2. Material and Methods

### 2.1. Subjects and Sampling

Ethics approval for this study was obtained from the Ethics Committee for Research Involving Human Subjects, Universiti Putra Malaysia (JKEUPM-2017-242). The trial was registered with the National Medical Research Register of Malaysia, NMRR-17-2640-38902. Sixty toddlers aged 1–4 years old were recruited from nurseries and daycare centers in different suburbs (Serdang, Kajang and Bangi) in Selangor, Malaysia. The children were screened based on their medical check-up records provided to us from child-care centres. Toddlers who were taking medication for chronic illnesses or not wearing diapers were excluded from the study. The children’s careers did not complete a food frequency questionnaire for the toddlers nor was anthropometric data on the recruited toddlers collected. Informed consent was obtained from each parent at the time of recruitment to the study. Parents of toddlers in all groups were asked not to provide any of the following to the infants one week before commencing the trial and throughout the trial period: marine origin food products (fresh fish, canned fish, fish paste, shellfish), sea vegetable (seaweed), walnut, avocado, egg yolks, omega-3 fortified products, chia seed, cod liver oil, fish oil supplements.

### 2.2. Intervention

In total, 60 toddlers were randomly allocated to four groups (*n* = 15; *N* = 60) based on the supplement(s) they received including: 1. unfortified toddler formula (control group), 2. unfortified toddler formula plus high DHA tuna oil (fish oil group), 3. fortified toddler formula with dairy-based microencapsulated high DHA tuna oil powder (Driphorm^®^ HiDHA^®^ 50) (ME formula 1 group), 4. fortified toddler formula with allergenic-free microencapsulated high DHA tuna oil powder (Driphorm^®^ HA HiDHA^®^ 30) (ME formula 2 group). The spray dried microencapsulated powders, Driphorm^®^ HiDHA^®^ 50 encapsulated with the Maillard Reaction Products (MRPs), which contains casein and glucose syrup and Driphorm^®^HA HiDHA^®^ 30 encapsulated with Octenyl Succinic Anhydride (OSA) which contains modified starch and carbohydrates, were used as the DHA source in this study. Tuna oil was used as the study oil, as it has the closest DHA:EPA ratio to human milk [49].

Refined tuna oil in the form of oil and microencapsulated powders with MRPs or OSA starch as encapsulants was provided by Nu-Mega Ingredients Pty Ltd. (Melbourne, VIC, Australia). Briefly, one mL of refined tuna oil contained 250 mg DHA and 50 mg EPA. Therefore, the daily dose for the fish oil treatment was set to one syringe containing 1 mL of tuna oil, delivering 250 mg DHA daily to the toddlers. Parents were provided with individually wrapped, N_2_ flushed 1 mL capped syringes, containing exactly 1 mL of tuna oil and asked to store them at room temperature. One gram of Driphorm^®^ HiDHA^®^ 50 powder contained 116 mg DHA plus 20 mg EPA and one gram of Driphorm^®^HA HiDHA^®^ 30 powder contained 70 mg DHA plus 10 mg EPA. In order to provide the equivalent dose of 250 mg of DHA per day as for the fish oil treatment, 2.2 g of microencapsulated tuna oil formula 1 powder and 3.3 g of microencapsulated tuna oil formula 2 powder were added into formula sachets for ME formula 1 and ME formula 2 groups respectively (Table 1). Parents were required to prepare the milk drink daily by measuring 200 mL of warm water into a bottle and then add the entire contents of the formula sachet (35 g) and shake (serving size: 228 mL). The toddler formula sachets were given to the parents on a weekly basis. Empty sachets and syringes were collected as a proof of consumption of provided toddler formulas and tuna oil.

All formula and syringe sachets were prepared by Nu-Mega Ingredient Pty Ltd., Brisbane, Australia. The total fat and nutrient composition of given formula sachets were determined by Dairy Technical Service Pty Ltd. (Melbourne, Australia). EPA and DHA analyses were conducted by Nu-Mega Ingredients Pty Ltd. (Melbourne, Australia). The microbiology quality (*Clostridium perfringens*, *Bacillus cereus, Salmonella* species, *Listeria* species, *Enterobacteriaceae* and *Enterobacter sakazakii)* of given samples was determined by Analytical Laboratory System (Melbourne, Australia).

### 2.3. Toddler Sleep and Cry Diary

Parents were asked to complete the Sleep and Cry Questionnaire for a period of three consecutive days; before, middle and after the intervention. For the crying section, time and duration (minute) of cry for three consecutive days were reported. For the sleep section, sleep and wake time (hour) during day and night were reported.

### 2.4. Blood Fatty Acid Measurement

Blood was collected from the thumb of toddlers using an automatic lancing device at the beginning (Day 0), middle (day 15) and the end of the intervention (day 30). The blood was spotted onto the collection area of a dried blood spot kit and air dried for 2 h at room temperature prior to storage at −20 °C. The samples were labelled with the participant’s identification number. All samples were then sent to Trajan Scientific and Medical (Adelaide, Australia) for fatty acid analysis and Omega-3 Index, at the end of the study. Omega-3 index is defined as the proportion of the sum of EPA and DHA content in the total fatty acid content in the erythrocyte and is expressed as a percentage [50].

Samples of faecal matter from soiled diapers were collected 3 times per week and stored in zip-lock plastic bags at −20 °C. For each subject, the faeces collected in one week were pooled and homogenised, and 1 g of pooled sample taken for the analysis (total weight of faeces not measured). The average of faeces data obtained in week one and two was considered middle point data, and the average of faeces data obtained in week three and four was considered end point data. The lipids were extracted by dichloromethane/methanol, using a modified method described by Folch et al. [51]. Fatty acids derived from the lipids were methylated using an acid-catalysed trans-methylation [52]. The fatty acid methyl esters were extracted into hexane and analysed by a gas chromatography system (Agilent Technologies, Santa Clara, CA, USA) equipped with a HP-88 capillary column (60 m × 0.25 mm) and using helium as the carrier gas. DHA and other fatty acid methyl esters were identified by comparison with the retention times of standards and were quantified on the basis of tricosanoic acid as the internal standard.

### 2.5. Data Analysis

Results were expressed as mean ± standard deviation (SD). The differences of the results between toddlers of different treatment were tested using two-way analysis of variance (ANOVA), assessing the effects of treatment, time and their interactions. Paired tests were performed with Tukey’s test at the significance level of 0.05.

## 3. Results

### 3.1. Subject Characteristics

A total of 60 Malaysian toddlers commenced the study, with 48 (*n* = 12/group) participating to completion; twelve toddlers withdrew for different reasons including hardening of their stools (*n* = 3), the onset of diarrhoea (*n* = 3) and a distaste for the formula (*n* = 6). None of the participants reported having any allergic reaction. The microbiological quality of the prepared sachet formulas was confirmed to meet specifications.

There were no significant differences in population characteristics among the four groups with the exception of gender. Of the 48 toddlers who completed the study, 52% were male and 48% female, however gender was not distributed equally among groups (Table 2). The mean age of toddlers completing the study was 2 years and 11 months (range 1–4 years), with no significant difference in age among the four groups (*p* = 0.169, Table 2).

### 3.2. Fatty Acid Levels in Whole Blood

For all 48 children who completed this study, the average percent of total saturated fatty acids in whole blood was 39.3%, followed by 31.9% of total *n*-6 PUFA, 23.6% of total monounsaturated fatty acids, and 4.7% of total *n*-3 PUFA. The main components of *n*-6 PUFA were linoleic acid (LNA, 18:2*n*-6) and arachidonic acid (ARA, 20:4*n*-6), accounting for 22.1% and 6.9% of whole blood, respectively. DHA was the main component of *n*-3 PUFA, accounting for 3.3% of total fatty acids in whole blood. Overall, the average percent of individual fatty acids (LNA, ARA, 20:3*n*-6 and 22:4*n*-6, EPA, DPA, DHA, and alpha-Linolenic acid (ALA, 18:3*n*-3) did not differ between groups at the commencement of the experiment.

Table 3 shows the mean whole blood fatty acid levels (% of total fatty acids) at three time points (baseline, middle and endpoint) in the different groups. In the control group, there was no significant difference in mean fatty acid levels at the endpoint compared with the middle point and the baseline. In the three treatment groups, all supplied with 250 mg DHA per day, there were significant differences in mean whole blood percent for most of omega-3 fatty acids at the end of the study (*p* < 0.05; Table 3). While mean levels of ALA and DPA remained unchanged after one month of treatment in all groups (Table 4), a significant difference in mean percentage of total omega-3 fatty acids EPA and DHA was observed after two weeks of supplementation with formulas containing microencapsulated fish oil powders (ME formula 1 and formula 2), but not with non-encapsulated fish oil (Table 4).

There was no significant difference between the Omega-3 Index of the four groups at the baseline with the exception of fish oil group. The overall mean of the Omega-3 Index for the 48 participating toddlers was 6.5%. This value falls into the ‘medium risk’ category based on the Index determined for cardiovascular mortality in adults (≤4% high risk, 4–8% medium risk, >8% low risk) [50]. Following one month of supplementation with toddler formula enriched with either high DHA fish oil or microencapsulated fish oil powders, the Omega-3 Index increased among children in the three treatment groups. As shown in Table 4, a significant (*p* < 0.05) increase in the mean Omega-3 Index was observed after two weeks supplementation with formulas containing microencapsulated fish oil powders (ME formula 1 and formula 2), but not with non-encapsulated fish oil supplementation.

### 3.3. Fatty Acid Levels in Faeces

As shown in Table 5, no significant differences were observed between the total amount of faecal fat in the groups over the 30-day study with the exception of ME formula 2 group. The average content of lipids in 1 g of faeces (± SD) at day 30 in all groups was between 6 mg to 10 mg and did not appear to be affected by diet. The content of excreted DHA (ng/g) was not significantly different between groups (Table 6). Fish oil supplementation was also associated with a small, but significant increase (*p* = 0.038) in the DHA content in faeces at the end point of the study, compared with baseline, but this was not significantly different from the microencapsulated tuna oil powders (ME formula 1 and formula 2).

### 3.4. Sleep and Cry Pattern

The children’s sleep and cry rates, as reported by their parents, are provided in Table 7. In this study, no significant effect of DHA supplementation on cry and sleep patterns was observed. The sleep duration (including naps) for toddlers was 11–12 h per day for all groups. Children went to bed between 7 p.m. and 9 p.m. and woke up around 6 a.m. and 8 a.m.

## 4. Discussion

Microencapsulated high DHA fish oil provides an effective way of delivering DHA and EPA, as it enables the incorporation of fish oil into a wide range of food products without affecting palatability and extends the stability of the oil. Several studies have compared the bioavailability of microencapsulated high EPA fish oil with non-encapsulated fish oil [42,43] or that which is encased in soft gel capsules [44,45,46,47,48]. These studies suggested that microencapsulation of high EPA fish oil increased the bioavailability of n-3 LC-PUFA in healthy adults to a similar [42,45,46,47,48] or greater [43,44] extent than non-encapsulated fish oil.

In the present study, there was a significant greater increase in the blood DHA when tuna oil was added to toddler formula as microencapsulated tuna oil powder compared with non-encapsulated tuna oil. Such data has not been reported previously in toddlers.

The higher blood DHA level might be due to the improved digestion and absorption of tuna oil due to smaller emulsion (oil droplet) size, and the effective protection of the oil encapsulant against the negative conditions (such as pH or the presence of pro-oxidising substances) in the gastrointestinal system of the toddlers. The mean oil droplet size in the emulsion of ME formula 1 and ME formula 2 was approximately 200 nm. Small oil droplet sizes present a greater surface area for the accessibility of lipases, bile salts and other biological surfactants that enable greater solubilisation and absorption rate [53]. Moreover, diminution of oil droplets in an emulsion matrix enhances lipolytic action of pancreatic lipases that are surface active at the interfacial [hydrophilic–hydrophobic] region. Micro-emulsions have also been reported to improve delivery and bioavailability of actives via ease of transportation through cell membranes and into the plasma [54], which could explain the increased blood DHA in toddler formulas with microencapsulated tuna oil powders. Lastly, the microencapsulated tuna oil was protected against the negative impact of other food ingredients, such as calcium ions present in milk, which might reduce the availability of *n*-3 LC-PUFA by forming a complex with free fatty acids [55,56].

An important understanding of the actual bioavailability of *n*-3 LC-PUFA is provided by measuring the fatty acid levels in faeces. In this study, the excretion of DHA was less than 1% of DHA intake (the mean weight of faeces from children aged 1–4 years has been reported as 85 g/day [57]) and no significant difference in DHA content in the faeces between groups was observed (Table 6).

Breast-feeding is recommended as the preferred choice for infant feeding. Yuhas et al. [58] determined the fatty acid composition of mature human milk (>four weeks of lactation), as well as maternal dietary intake, from approximately 50 women across nine countries (Australia, Canada, Chile, China, Japan, Mexico, Philippines, UK, and USA). DHA levels varied from 0.17% in Canada and the USA, to 0.99% in Japan, and were found to be related to the maternal dietary intake of fish. In contrast, ARA levels ranged from 0.36% in the UK to 0.49% in China, with no apparent relationship to dietary intake. In another study, Brenna et al. [59] reviewed data from 65 studies (2474 women) and reported that the breast milk level of DHA was 0.32 ± 0.22% (mean ± SD) and ARA was 0.47 ± 0.13% worldwide. A more recent survey [60] indicated marginally higher levels of DHA (0.37 ± 0.11%) and ARA (0.55 ± 0.14%).

DHA supplementation during lactation has been shown to substantially increase breast milk DHA content [61]. Based on an assessment of the maternal diet and the effect of diet on breast milk composition, Jackson and Harris [62] recommended a DHA target of 0.3% for human milk DHA. In order to achieve this, breast-feeding women should aim for a minimum average daily supply of 200 mg DHA [63]. While a DHA target of 0.3% for human milk is an appropriate reference point for consideration in infant formulas, clinical studies have evaluated term infant formulas containing as little as 0.1% [64] and as high as 0.96% DHA [65,66,67].

Of the available studies on infant/toddler formula composition, there is little information available on DHA content and, where reported, levels vary substantially. Riva et al. [68] analysed the composition of 30 different infant formulas available on the Italian market and identified a variation between 4.32 to 5 g/100 kcal for the lipid content. Only four of the infant formulas analysed contained DHA, with levels varying between 17 to 25 mg/100 kcal. Zunin et al. [69] also analysed 32 infant formulas marketed in Italy and identified a variation between 22.9 and 30.1 g/100 g (dry basis) for the total lipid fraction. Only 3 of these formulas contained n-3 LC-PUFA, all with a higher DHA to EPA ratio (level of DHA not specified). Rêgo et al. [70] performed an assessment of 87 infant formulas available on the Portuguese market and identified that the total lipid fraction ranged from 4.7 to 5.3 g/100 kcal, the closest of the reported levels to the human milk reference value of 5.6 g/100 kcal, while Mendonça et al. [71] determined the lipid content in 10 infant formula samples from Brazil market and found that the lipid fraction varied between 2.60 and 4.27 g/100 g; DHA was not quantified in these studies. Several studies examined US marketed infant formula with 0.32% fatty acids from DHA (17 mg/100 kcal) in comparison with experimental formulas in studies examining visual acuity, including 0.64% DHA (34 mg/100 kcal) or 0.96% DHA (51 mg/100 kcal) [72,73]. The results of these studies indicated that supplementation of infant formula with 0.32–0.64% DHA was sufficient to promote visual acuity maturation during infancy.

In an early recommendation by the European Society for Pediatric Gastroenterology Hepatology and Nutrition (ESPGHAN), the total lipid content in infant formulas was advised to be between 4.4 and 6 g/100 kcal and, together with the optional addition of DHA, where should not exceed a total fat intake of 0.5% [74]. Following a more recent recommendation on the essential composition of infant and follow-on formula, the mandatory addition of DHA to infant and follow-on formula is a new requirement introduced by the European Union [75]. Based on this regulation, DHA has become a mandatory ingredient at a minimum of 20 mg/100 kcal and a maximum of 50 mg/100 kcal for infant (0–12 months) formula, and ARA is an optional ingredient. With the European Union enforcing mandatory fortification of infant formula by February 2020, stable and bioactive high DHA powders provide manufacturers with a viable solution for increasing the DHA dosage levels in their existing formula products, without compromising the stability and flavour. Despite the availability of enriched DHA formula in the European and Australian markets, the levels are still lower in composition compared with human milk; the fortification level was around 38–90 mg of DHA for daily intake, with only premium brand products providing fortification levels at around 100 mg [76].

While differences in the composition of formula exist, studies have shown the positive effects of omega-3 fatty acids at varying levels. In the present study, the level of total blood *n*-3 PUFAs in Malaysian toddlers increased significantly after one month of supplementation with fish oil and microencapsulated tuna oil powders (Table 3). There have been no published data on the dietary intake of *n*-3 fatty acids in Malaysian toddlers, however data from USA [20] and Japan [24] indicate intakes of total *n*-3 PUFA in this age group to be between 0.85 to 1.1g/day, respectively, with most of the *n*-3 from alpha-linolenic acid. Compared with US recommendations, the intakes of total *n*-3 and *n*-6 fatty acids in the present study were lower, whereas the *n*-6:*n*-3 ratio was similar [77]; at the end of the study, the *n*-6:*n*-3 ratios were 5.80 ± 0.93, 4.93 ± 1.32 and 5.24 ± 1.12 in fish oil, ME formula 1 and ME formula 2 groups, respectively. It has been reported that an *n*-6:*n*-3 ratio below 4:1 is likely to result in a markedly different fatty acid composition from that present in human milk [78]. Therefore, the recommended ranges of between 5:1–15:1 and 6:1–16:1 for the intake of *n*-6:*n*-3 have been established for infant formulas and follow-up formulas [79,80]. Although in the present study, the Malaysian toddler intake *n*-6:*n*-3 ratio was similar to that of toddlers in the US [20], it was higher than those reported for toddlers in the UK [19], Japan [24], and China [22].

Omega-3 Index is an indicator of long-term bioavailability as well as a good indicator of the incorporation of fatty acids into tissues [55]. As reported by Harris and Von Schacky [81], Omega-3 Index values of >8% are associated with the greatest cardio-protection, whereas index values of <4% are associated with the least. The average Omega-3 Index among Malaysian toddlers was 6.5%, which is higher than that of school-age Australian children between 5–12 years (5%) [82]. However, if the Omega-3 Index values apply to children, the data for children in both Malaysia and Australia fall into the ‘medium risk’ category of the Omega-3 Index. In the present study, a significant increase in the Omega-3 Index was observed after just two weeks of supplementation with microencapsulated fish oil powders (ME formula 1 and 2). The Omega-3 Index was 8.8% and 8.6% for toddlers in the ME formula 1 and ME formula 2 groups, respectively, compared with 6.8% in the non-encapsulated fish oil group at the end of the study.

In the present study, sleep duration or disturbance was not associated with the blood fatty acid profiles in the toddlers; all 48 toddlers slept 11–12 h per day (including naps) regardless of the treatment, and no effect was found after one month of supplementation with approximately 250 mg DHA per day. There is some evidence to support the role of these fatty acids in sleep regulation [83,84], however, this has received little investigation in adults, and even less so in toddlers and school-aged children. Montgomery et al. [29] showed that DHA supplementation in children led to increased sleep duration and fewer waking episodes per night, while in another study, dietary supplementation with omega-3 DHA for 16 weeks improved parent-rated sleep in children [28]. Frequent fish eating has also been found to be independently associated with less sleep disturbances, indicating better overall sleep quality [30]; children who often or sometimes consumed fish had higher total sleep disturbance scores of 4.49 (*p*  =  0.001; Cohen’s *d*  =  0.221) and 3.01 (*p*  =  0.019; Cohen’s *d*  =  0.132), respectively, than those who never or seldom ate fish.

We acknowledge that the results of this study are limited by the small sample size. Furthermore, the total weight of faeces was not measured, and would be useful to investigate in further studies to determine the excretion of DHA; however, in this study, the estimated excretion of DHA was very low compared with intake (less than 1%). In terms of sample preparation, DHA in ME formulas 1 and 2 were measured accurately, however, DHA in the fish oil group was provided as tuna oil in syringes and this was added, by the parents, to the milk drink given to their children. As such, the actual quantity of DHA provided might have varied due to delivery of the oil from the syringes by the individual parents. Although all subjects were allocated to different groups at random, the parents of the subjects in the fish oil group were not blinded since they were adding fish oil to the milk drink, in comparison with the other groups, who simply dissolved the contents of the sachets in water. This was, however, not expected to have a significant impact on the results. Furthermore, children were screened based on their medical check-up records provided to us from child-care centres, but anthropometric data was not collected, which is another limitation of the study. We also acknowledge that the lack of omega 3 dietary intake data for the toddlers prior to the start of the study is a limitation of this study, however, the baseline blood *n*-3 values were judged to be in a normal range, showing no evidence of biochemical *n*-3 deficiency. Finally, in this study the entire collected blood spot was used for fatty acid analyses so there was no additional blood for other analyses.

A strength of the study was that it was the first to assess the bioavailability of *n*-3 DHA from infant formulas for toddlers (in Malaysia). Studying toddlers in this age group is challenging for several reasons, including being able to take blood samples. In this study, use of finger prick blood sampling was employed, which is a minimally invasive technique.

## 5. Conclusions

Together, the data from this study showed significant differences in the bioavailability of DHA from microencapsulated formulas compared with fish oil in a one-month study in toddlers. Using microencapsulated tuna oil powders is an effective approach to incorporate bioavailable DHA into infant and toddler formulas. The results from this study provide a suitable platform for future human studies aimed at developing substantiated evidence for advising consumers on the most efficient way to increase their *n*-3 LC-PUFA status.

## Figures and Tables

**Table 1 nutrients-12-00248-t001:** Nutrition Information of sachets.

	Control	Fish Oil	ME Formula 1	ME Formula 2
Label	Sachet 1	Sachet 1	Sachet 2	Sachet 3
Serving size (g)	35	35	35	35
Nutrient compositions ^1^/serve ^2^				
Energy (KJ)	645	645	670	670
Protein (g)	6.9	6.9	6.5	6.5
Carbohydrate (g)	19.6	19.6	20.0	20.1
Fat–Total (g)	5.9	5.9	5.9	5.9
Saturated (g)	2.5	2.5	2.6	2.6
Monounsaturated Fats (g)	−	−	1.4	1.4
Polyunsaturated Fats (g)	−	−	1.8	1.8
DHA (mg)	−	−	250.2	248.7
EPA (mg)	−	−	65	62
Microencapsulated fish oil powder ^3^	−	−	+	+
Fish oil ^4^	−	+	−	−
DHA (mg)	−	250	−	−
EPA (mg)	−	50	−	−

DHA: docosahexaenoic acid; EPA: eicosapentaenoic acid; ME: Microencapsulated. ^1^ Average data from three samples. ^2^ Serving size is 228 mL of milk drink (35 g of powder into 200 mL of warm water). ^3^ Microencapsulated fish oil powders were already mixed in the sachets. ^4^ Fish oil provided in individual 1 mL syringes and packed under N_2_ gas. Nutrient compositions of unfortified milk powders (sachet 1) were reported directly from the toddler formula tin. Nutrient composition of ME formula 1 (sachet 2) and ME formula 2 (sachet 3) was determined by Dairy Technical Service Pty Ltd. (Melbourne, Australia).

**Table 2 nutrients-12-00248-t002:** Participant characteristics.

Time/Group	Control (*n* = 12)	Fish Oil (*n* = 12)	ME Formula 1 (*n* = 12)	ME Formula 2 (*n* = 12)	*p* Value
Average age (month)	37 ± 10	35 ± 11	31 ± 11	40 ± 7	0.169
Gender (% of Male)	50	31	70	57	

**Table 3 nutrients-12-00248-t003:** Mean whole blood fatty acid levels (%) in toddlers fed in toddlers fed different formulas.

	Control			Fish Oil			ME Formula ^1^			ME Formula ^2^			*p* _(group)_	*p* _(Time)_	*p* _(interaction)_
	Baseline ^1^	Middle Point ^2^	Endpoint ^3^	Baseline	Middle Point	Endpoint	Baseline	Middle Point	Endpoint	Baseline	Middle Point	Endpoint			
Total saturated	39.25 ± 2.33	38.34 ± 1.48	38.16 ± 1.65	39.79 ± 2.69	39.10 ± 2.42	39.01 ± 2.27	38.99 ± 2.53	37.54 ± 1.75	38.14 ± 1.95	39.46 ± 1.91	38.73 ± 1.26	38.45 ± 1.55	0.129	0.270	0.989
Total monounsaturated	23.27 ± 1.38	23.64 ± 1.79	23.82 ± 1.48	23.02 ± 1.73	23.73 ± 2.33	22.94 ± 2.39	24.33 ± 3.14	23.73 ± 2.63	23.27 ± 1.95	24.16 ± 2.95	22.56 ± 0.92	21.58 ± 1.77	0.162	0.165	0.200
Total *n*-9 PUFA	20.46 ± 1.40	20.79 ± 1.58	20.98 ± 1.51 ^a^	20.29 ± 1.58	20.99 ± 2.43	20.41 ± 2.48 ^a^	21.61 ± 2.86	20.82 ± 2.75	20.74 ± 1.86 ^a^	21.33 ± 2.45	19.87 ± 1.09	18.79 ± 1.45 ^b^	0.134	0.220	0.132
Total *n*-7 PUFA	2.25 ± 0.42	2.36 ± 0.41	2.28 ± 0.36	2.21 ± 0.36	2.07 ± 0.30	2.02 ± 0.31	2.25 ± 0.41	2.29 ± 0.43	2.09 ± 0.33	2.37 ± 0.89	2.17 ± 0.41	2.11 ± 0.58	0.332	0.292	0.841
Total *n*-6 PUFA	32.21 ± 2.17	32.76 ± 2.33	32.79 ± 2.02	32.40 ± 2.35	31.68 ± 2.53	32.06 ± 2.38	31.58 ± 3.74	31.87 ± 2.81	31.55 ± 2.42	31.51 ± 4.02	32.07 ± 1.52	33.13 ± 2.32	0.505	0.697	0.845
Total *n*-3 PUFA	5.08 ± 1.30 ^a^	4.95 ± 1.12 ^a^	4.67± 0.63 ^a^	4.47 ± 0.81 ^A^ ^b^	5.10 ± 1.07 ^A^ ^ab^	5.65 ± 0.96 ^B^ ^b^	4.89 ± 0.59 ^A^ ^b^	6.54 ± 1.28 ^B^ ^b^	6.75 ± 1.51 ^C^ ^c^	4.62 ± 0.88 ^A b^	6.35 ± 1.38 ^B b^	6.56 ± 1.28 ^C c^	<0.001	<0.001	0.001

Value expressed as mean ± SD. In each row, values with lowercase different letters differ significantly for group and values with uppercase different letters differ significantly for time (*p* < 0.05). ME: Microencapsulated; *n*-6 PUFA: 18:2*n*-6, 20:2*n-*6, 22:2*n*-6, 18:3*n*-6, 20:3*n*-6, 20:4*n*-6, 22:4*n*-6, 24:4*n*-6, 24:5*n*-6, 22:5*n*-6; *n*-3 PUFA: 18:3*n*3, 20:3*n*-3, 22:3*n*-3, 18:4*n*-3, 20:4*n*-3, 20:5*n*-3, 22:5*n*-3, 24:5*n*-3, 24:6*n*-3, 22:6*n*-3. ^1^ Data from day 0. ^2^ Data from day 15. ^3^ Data from day 30.

**Table 4 nutrients-12-00248-t004:** Selected whole blood omega-3 fatty acids (%) and Omega-3 Index in toddlers fed different formulas.

	Control			Fish Oil			ME Formula ^1^			ME Formula ^2^			*p* _(group)_	*p* _(Time)_	*p* _(interaction)_
	Baseline ^1^	Middle Point ^2^	Endpoint ^3^	Baseline	Middle Point	Endpoint	Baseline	Middle Point	Endpoint	Baseline	Middle Point	Endpoint			
*n*-6 PUFA															
LNA (18:2*n*-6)	22.02 ± 2.15	22.82 ± 2.20	22.80 ± 2.01	22.21 ± 1.73	22.79 ± 1.74	23.13 ± 1.75	22.25 ± 2.83	22.80 ± 2.25	22.46 ± 2.39	21.85 ± 3.48	22.76 ± 2.08	23.71 ± 2.35	0.940	0.096	0.922
20:3*n*-6	1.25 ± 0.24	1.32 ± 0.15	1.30 ± 0.21	1.34 ± 0.21 ^A^	1.04 ± 0.18 ^B^	1.08 ± 0.20 ^AB^	1.25 ± 0.31	1.14 ± 0.26	1.10 ± 0.21	1.24 ± 0.26	1.14 ± 0.21	1.13 ± 0.20	0.202	0.011	0.132
ARA (20:4*n*-6)	7.58 ± 0.81 ^a^	7.27 ± 1.06	7.31 ± 0.93	7.40 ± 0.99 ^ab^	6.64 ± 1.13	6.60 ± 1.11	6.75 ± 1.14 ^ab^	6.70 ± 1.36	6.80 ± 0.97	7.08 ± 1.11 ^b^	6.965 ± 0.85	7.08 ± 0.80	0.045	0.242	0.789
22:4*n*-6	0.98 ± 0.18	0.96 ± 0.18	0.98 ± 0.24	1.06 ±0.21	0.90 ± 0.23	0.91 ± 0.18	0.93 ± 0.23	0.88 ± 0.21	0.84 ± 0.19	0.97 ± 0.21	0.89 ± 0.23	0.89 ± 0.16	0.278	0.093	0.877
*n*-3 PUFA															
ALA (18:3*n*-3)	0.29 ± 0.11 ^b^	0.39 ± 0.16 ^ab^	0.33 ± 0.13 ^ab^	0.28 ± 0.11 ^b^	0.39 ± 0.18 ^ab^	0.37 ± 0.16 ^ab^	0.39 ± 0.15 ^a^	0.48 ± 0.24 ^a^	0.41 ± 0.18 ^a^	0.26 ± 0.09 ^b^	0.32 ± 0.15 ^b^	0.27 ± 0.11 ^b^	0.001	0.008	0.963
EPA (20:5*n*-3)	0.45 ± 0.25	0.39 ± 0.12 ^a^	0.45 ± 0.25 ^a^	0.33 ±0.11 ^A^	0.52 ± 0.19 ^B^ ^ab^	0.61 ± 0.21 ^B^ ^b^	0.41 ± 0.15 ^A^	0.73 ± 0.35 ^B bc^	0.72 ± 0.26 ^B bc^	0.38 ± 0.18 ^A^	0.87 ± 0.46 ^B c^	0.82 ± 0.30 ^B c^	<0.001	<0.001	0.002
DPA (22:5*n*-3)	0.72 ± 0.20	0.74 ± 0.18	0.75 ± 0.19	0.77 ± 0.28	0.69 ± 0.24	0.76 ± 0.22	0.71 ± 0.19	0.85 ± 0.43	0.72 ± 0.17	0.69 ± 0.24	0.88 ± 0.36	0.81 ± 0.36	0.715	0.521	0.552
DHA (22:6*n*-3)	3.61 ± 0.89 ^a^	3.37 ± 0.73 ^a^	3.22 ± 0.40 ^a^	3.09 ± 0.62 ^A b^	3.52 ± 0.76 ^AB a^	3.91 ± 0.65 ^B b^	3.37 ± 0.45 ^A ab^	4.46 ± 0.71 ^B b^	4.90 ± 1.16 ^B c^	3.27 ± 0.64 ^A ab^	4.34 ± 0.75 ^B b^	4.58 ± 0.76 ^B c^	<0.001	<0.001	<0.001
Omega-3 Index	6.94 ± 1.36 ^a^	6.62 ± 1.22 ^a^	6.41 ± 0.61 ^a^	6.10 ± 0.91 ^A b^	6.89 ± 1.22 ^AB a^	7.54 ± 1.07 ^B b^	6.60 ± 0.68 ^A ab^	8.39 ± 1.25 ^B b^	8.89 ± 1.66 ^B c^	6.42 ± 0.94 ^A ab^	8.65 ± 1.27 ^B b^	8.41 ± 1.38 ^B c^	<0.001	<0.001	<0.001
*n*-6:*n*-3	6.65 ± 0.35	6.92 ± 0.36 ^a^	6.90 ± 0.36 ^a^	7.45 ± 0.36 ^A^	6.47 ± 0.36 ^B b^	5.80 ± 0.37 ^B b^	6.57 ± 0.32 ^A^	5.05 ± 0.36 ^B c^	4.93 ± 0.36 ^B c^	7.02 ± 0.35 ^A^	5.30 ± 0.30 ^B bc^	5.24 ± 0.25 ^B bc^	<0.001	<0.001	0.029

ME: Microencapsulated; LNA: linoleic acid; ARA: arachidonic acid; ALA: alpha-Linolenic acid; EPA: eicosapentaenoic acid; DPA: docosapentaenoic acid; DHA: docosahexaenoic acid. Value expressed as mean ± SD. In each row, values with lowercase different letters differ significantly for group and values with uppercase different letters differ significantly for time (*p* < 0.05). All active groups were provided 250 mg DHA per day. ^1^ Data from day 0. ^2^ Data from day 15. ^3^ Data from day 30.

**Table 5 nutrients-12-00248-t005:** Total amount of faecal fat (mg/g wet weight) in all groups over the 4 weeks.

Time/Group	Control	Fish Oil	ME Formula 1	ME Formula 2	*p* Value
Baseline ^1^	8.23 ± 4.55	10.10 ± 6.84	8.24 ± 2.95	10.00 ± 5.79 ^a^	0.676
Middle point ^2^	6.54 ± 3.99	7.04 ± 3.24	7.88 ± 4.17	6.65 ± 2.53 ^b^	0.766
Endpoint ^3^	6.93 ± 2.67	6.37 ± 1.92	6.69 ± 2.45	6.69 ± 1.78 ^b^	0.945
*p* value	0.503	0.122	0.447	0.047	

Values are mg lipids in 1 g faeces weight (wet) ± standard deviation, *n* = 12 per group. Values with different letters in each column differ significantly (*p* < 0.05). There are no significant differences between groups and experimental days (except for ME formula 2). ^1^ Data from day 0. ^2^ Average data from week 1 and 2. ^3^ Average data from week 3 and 4. ME: Microencapsulated.

**Table 6 nutrients-12-00248-t006:** DHA content of toddler faeces (ng/g wet weight) in all groups over the 4 weeks.

Time/Group	Control	Fish Oil	ME Formula 1	ME Formula 2	*p* Value
Baseline ^1^	10.5 ± 20.7	<0.1	2.5 ± 9.2	2.9 ± 1.0	0.186
Middle point ^2^	3.1 ± 7.8	<0.1	<0.1	1.2 ± 4.3	0.265
Endpoint ^3^	2.7 ± 6.6	4.1 ± 7.4	3.1 ± 7.6	2.7 ± 6.1	0.794
*p* value	0.259	0.038	0.789	0.553	

Results are expressed as mean ± SD, *n* = 12 per group. There are no significant differences between groups and experimental days (with the exception of fish oil group). ^1^ Data from day 0. ^2^ Average data from week 1 and 2. ^3^ Average data from week 3 and 4. ME: Microencapsulated.

**Table 7 nutrients-12-00248-t007:** Total minutes of cry and total hours of sleep by different groups.

Time/Group	Control	Fish Oil	ME Formula 1	ME Formula 2	*p* Value
Cry pattern (minutes of crying/day of record)					
Baseline ^1^	0.79 ± 2.7	2.25 ± 4.13	2.53 ± 3.55	2.31 ± 4.83	0.653
Middle point ^2^	1.15 ± 4.12	2.08 ± 3.96	2.38 ± 7.79	1.15 ± 2.99	0.321
Endpoint ^3^	0.84 ± 1.94	1.63 ± 3.26	1.53 ± 5.54	1.53 ± 3.15	0.566
*p* value	0.878	0.466	0.198	0.729	
Sleep pattern (hours of sleep/day of record)					
Baseline	11.24 ± 1.55	11.62 ± 1.42	11.53 ± 1.57	10.64 ± 1.36	0.274
Middle point	12.01 ± 1.33	11.61 ± 1.53	11.74 ± 1.41	11.53 ± 0.86	0.588
Endpoint	11.89 ± 1.09	11.68 ± 1.02	11.89 ± 1.37	10.90 ± 1.17	0.843
*p* value	0.618	0.859	0.430	0.772	

Results are expressed as mean ± SD (*n* = 12 per group). ^1^ Data from day 0. ^2^ Data from day 15. ^3^ Data from day 30. ME: Microencapsulated.

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
