# Peer review of "Microencapsulated Tuna Oil Results in Higher Absorption of DHA in Toddlers"

_nutrients, 2020, doi:10.3390/nu12010248_

Round 1

Reviewer 1 Report

Manuscript ID: nutrients-668067: Microencapsulated Tuna Oil Results in Higher Absorption of DHA in Toddlers

The article address an important field of clinical research, aiming to develop correct nutritional intake of DHA in toddlers through milk formulas or supplementation.

The text is fluent and accurate, despite an intrinsic complexity of the field, might be implemented in discussion in some points:

I would appreciate a comment of Authors on applicability of Omega-3 Index in childhood If the authors consider the DHA micro-encapsulated formulation more rapid, but comparable to a traditional fish oil in a longer time, as the levels reached after a month of administration suggest

Author Response

Please see word attachment.

PLEASE NOTE LINE NUMBERS HAVE CHANGED FROM ORIGINAL SUBMITTED VERSION

Reviewer 2 Report

The research of Fard et al. provides proper data regarding to the significant increase found in the plasma DHA when tuna oil was added to toddler formula as microencapsulated tuna oil powder compared with non-encapsulated tuna oil, which has not been reported before. However, there are major issues that should be considered and corrected / improved regarding to the study design and methods.

The absorption and bioavailability of a supplement (DHA and EPA in this case) can vary a lot, and therefore, have a completely different effect if the supplementation has been done in subjects with no deficiency (added to an appropriate level to improve the health) or in subjects with low levels (covering a deficiency to avoid imbalance). Thus, studies testing the effect of a concrete supplement should be performed not only taking into account the amounts throughout the trial period or a week before the study, but also they should consider the habitual diet overall before the trial to make sure whether toddlers were consuming accurate amounts of omega-3 on their regular dietary pattern.

Even if the parents of toddlers in all groups were asked not to provide omega-3 fortified products, fish supplements or foods containing omega-3 to the infants one week before commencing the trial and throughout the trial period, there is the need to measure the regular consumption of the toddlers (by asking to the parents) with 24-hour food recall together with a food frequency questionnaire with an emphasis on foods containing / rich in omega-3 PUFA.

Although authors acknowledge that subjects in the fish oil group were not blinded and this was not expected to have a significant impact on the results, for this kind of clinical trials a double-blind control is the suitable design model to avoid any expectation / conflict during sample analysis.

In PAGE 1 – lines 31-32, authors mention fish oils originating from anchovies or the tuna oil used in the present study. However, they do not mention the salmon, which has the highest amount of omega-3 among oily fish species. In this sense, salmon oil contains (for 5mL recommended diary dose): 550mg DHA + 425mg EPA, therefore salmon oil should worth considering, even as a diet supplement model in the present study.

In PAGE 1 – lines 43-44, authors mention that the current average n-3 LC-PUFA and estimated DHA intakes in toddlers in most countries are lower than the recommended levels and they add a Supplementary Table to show this. However, these data do not show the situation in Malaysia where the study has been conducted with 60 Malaysian toddlers (page 5, line 174). Besides, and as mentioned above, authors do not provide any information on the regular omega-3 consumption for the toddlers participating in the trial either.

In PAGE 3 – line 97, authors mention that exclusion criteria were taking medication for chronic illnesses or not wearing diaper. According to authors the study is performed in healthy toddles, however they do not provide enough data to this regard, e.g. basic blood biochemical parameter analysis, weigh, height or BMI of the toddlers.

In PAGE 3 – line 126-127 authors mention that empty sachets and syringes were collected as a proof of consumption of provided toddler formulas and tuna oil, however this is a proof of use not consumption per se.  Given that samples were collected at the beginning (Day 0), middle (day 15) and the end of the intervention (day 30), they could have used these sessions to take a control and make sure that the toddlers in each group were consuming the right amounts properly distributed.

Given the challenges when working with infant samples, the authors could have taken advantage of the samples already collected to improve the quality of the results and the study overall.

For example, authors acknowledge that the total weight of faeces was not measured, and would be useful to investigate in further studies to determine the excretion of DHA, however it would seem that the excretion of DHA was very low compared with intake. Given that no significant difference in DHA content in the faeces between groups was observed, authors could have done further / deeper analysis.

Author Response

Please see attached word document.

PLEASE NOTE LINE NUMBERS HAVE CHANGED FROM ORIGINAL SUBMITTED VERSION

Round 2

Reviewer 2 Report

The article has improved in its overall quality:

L97-98 Methods Included Trial Registration details: this can be taken as an evidence for the adequacy on the study Design and Methods.

Authors have also acknowledged and explained the main limitations in the study.

L119-120 Comment on choice of tuna oil: “Tuna oil was used as the study oil, as it has the closest DHA:EPA 119 ratio to human milk.” PLEASE, ADD REFERENCE

Author Response

Thank you for your positive comments.

We added a reference which shows that the DHA to EPA ratio in tuna oil is very similar to human milk (approx 4.2 to 1).